# Comparing dual oral agents plus insulin vs. Triple oral agents in uncontrolled type II diabetes: A pilot study

Nadia Gul[1], Inayat Ur Rehman[1,2], Yasar shah[1], Arbab Muhammad Ali[3]*, Zahid Ali[4], Omer Shehzad[1], Khang Wen Goh[5], Long Chiau Ming[6]*, Amal K. Suleiman[7]

1 Department of Pharmacy, Garden Campus, Abdul Wali Khan University Mardan, Mardan, Pakistan, 2 Department of Clinical Pharmacy and Pharmacy Practice, Faculty of Pharmacy, Universiti Malaya, Kuala Lumpur, Malaysia, 3 Department of Nephrology, Lady Reading Hospital Peshawar, Peshawar, Pakistan, 4 Department of Pharmacy, University of Peshawar, Peshawar, Pakistan, 5 Faculty of Data Science and Information Technology, INTI International University, Nilai, Malaysia, 6 School of Medical and Life Sciences, Sunway University, Bandar Sunway, Malaysia, 7 Department of Pharmacy Practice, College of Clinical Pharmacy, King Faisal University, Al Ahsa, Saudi Arabia

* drarbab_ali1@yahoo.com (AMA); chiaumingl@sunway.edu.my (LCM)

**Data Availability Statement:** At the time of data collection and obtaining permission to communicate with patients for this research, the data owner, Abdul Wali Khan University Mardan

## Abstract

### Introduction

Type II Diabetes mellitus (T2DM) patients often do not achieve glycemic control with oral hypoglycemic agents (OHAs). There are two main approaches to address this challenge: transitioning to a triple OHA regimen, or adding Insulin to the existing dual OHA regimen.

### Aim

This study aimed to compare the efficacy of adding Insulin to dual OHAs (Sitagliptin + Metformin) against adding a third OHA to Sitagliptin + Metformin in achieving glycemic control among patients with uncontrolled T2DM.

### Method

A pre-post study was conducted between 21 September 2023 and 21 December 2023 at Services Hospital Peshawar, Pakistan. Patients with uncontrolled T2DM with >7% HbA1c were divided into group 1 (Sitagliptin + Metformin plus a third OHA), and group 2 (Sitagliptin + Metformin plus pre-mixed Insulin 70/30). Glycemic control based on HbA1c values, fasting and random blood sugar levels, lipid profile, and body weight were evaluated after 3 months of therapy. The pre- and post- effect was compared by using a paired t-test.

### Results

The study included n = 80 patients with T2DM. Between groups 1 and 2, no significant difference was found in HbA1c values (9.1 vs. 9, with p = 0.724). However, BMI, cholesterol, and LDL significantly decreased in group 1 compared to group 2 (p<0.001 vs. p = 0.131, p = 0.023 vs. p = 0.896, and p = 0.003 vs. p = 0.395, respectively). Additionally, the incidence of hypoglycemic episodes was significantly lower in group 1 (7.5%) than in group 2 (47.5%, p =

(AWKUM), established stringent guidelines to ensure the confidentiality and privacy of patient information are maintained. Therefore the data may not be shared publicly. Data will be provided upon reasonable request if the requestor provides a valid justification for access. Data access requests may be sent to Prof. Dr. Haroon Khan, Chairperson of the Department of Pharmacy, AWKUM, at chairmanpharmacy@awkum.edu.pk.

**Funding:** The author(s) received no specific funding for this work.

**Competing interests:** All authors declare no financial and conflict of interest.

0.004). No significant difference was observed between the triple OHA and dual OHA plus Insulin regimens in achieving glycemic control.

## Conclusion

The triple OHA regimen improved BMI, cholesterol, and LDL levels, and reduced hypoglycemic episodes more effectively than dual OHA plus Insulin, despite similar HbA1c outcomes, suggesting it may be preferable for uncontrolled T2DM.

## Introduction

Diabetes mellitus (DM) is a serious, long-term condition with a major impact on the lives and well-being of individuals, families, and societies worldwide [1]. In 2021, diabetes affected around 537 million adults globally between the ages of 20 to 79 [2]. Due to its accelerated prevalence, it is projected that this number may rise to 783.2 million in 2045 [3]. A total of 6.7 million deaths were attributed to DM in 2021 [4]. The increasing trend of T2DM is the result of ageing [5], a rapid increase in urbanization [6], obesogenic environments [7], sedentary lifestyles [8], an increasing trend in Type 1 DM worldwide as well [9, 10], and genetic susceptibility [11]. Nearly 80% of people with T2DM are living in low- and middle-income countries [12].

According to the International Diabetes Federation, the Asian population has a significantly higher risk of developing diabetes and its related complications [13]. About 80% of patients that contribute to the global diabetes burden come from low- and middle-income countries, and of the total, more than 60% live in Asia [14]. Pakistan is ranked no 1 in the world in its proportion of diabetic patients, with a prevalence rate of 30.8% [15]. According to a 2019 study, the prevalence of T2DM was 17.1% [16]. The prevalence was higher in men (11.20%) than women (9.19%), and it was more common in cities than in rural areas [17]. Different factors are contributing to the increased prevalence of T2DM in Pakistan: a carbohydrate-rich diet and physical inactivity [18], a rapid increase in urbanization [6], no formal education, and obesity [19]. In Pakistan, the majority of patients with T2DM remain undiagnosed [19]. DM and hypertension are considered the major and leading causes of chronic kidney disease (CKD) [20, 21].

T2DM is a metabolic disorder affecting multiple organs, with complications more prevalent in uncontrolled cases, contributing to morbidity and mortality [22, 23]. Complications are classified as microvascular, including neuropathy [24], nephropathy [25], and retinopathy [26] and microvascular complications, such as stroke [24], cardiovascular disease (CVD) [27], and peripheral vascular disease (PVD) [28] are macrovascular complications. PVD may lead to bruises or injuries not healing over time, which can lead to gangrene and amputation [22].

Diabetes mellitus can be controlled by using pharmacological and non-pharmacological interventions [29]. Blood pressure, lipid profile, body weight, and HbA1c are essential in the clinical management of T2DM patients [30–32]. HbA1c is a marker of the average glucose levels spread over a two- to three-month period [33]. Another study also confirms the effectiveness of diabetes mellitus self-care management, showing improvement in metabolic markers like blood pressure, HbA1c, total cholesterol, high-density lipoprotein (HDL), and low-density lipoprotein (LDL) [34]. As the first-line treatment for T2DM, an OHA monotherapy is recommended [35]. Initiation of combination therapy has been proposed as an approach to maintain target blood glucose levels in T2DM patients [36, 37]. Previous studies demonstrate that in the

initial management of diabetes, Metformin and sulfonylureas were frequently used agents in dual therapy [38–41], whereas thiazolidinediones and dipeptidyl peptidase-4 inhibitors were considered as other treatment option in cases when dual therapy was not effective [40, 42, 43]. Clinical research has shown that combination therapy's synergy is safer and more effective than monotherapy [44].

At present, the literature lacks substantial clinical evidence regarding treatment decisions for patients with uncontrolled T2DM treated with dual oral OHA combinations. In order to effectively achieve and maintain glycemic control, the majority of T2DM patients will require Insulin, either alone or in combination with other OHAs [45, 46]. Considering the heightened prevalence of DM in Pakistan, along with a substantial number of patients experiencing uncontrolled DM, a significant portion of patients require triple therapy. Hence, this study aimed to compare the efficacy of adding Insulin to dual OHAs (Sitagliptin + Metformin) against adding a third OHA to Sitagliptin + Metformin in achieving glycemic control among patients with uncontrolled T2DM.

## Method

### Study design and setting

A pre- and post- study design was used for this study. The data was collected from 21 September 2023 to 21 December 2023 to compare the effectiveness of dual OHAs with additional Insulin as compared to triple OHAs for optimum control of HbA1c among uncontrolled T2DM patients. This study was conducted in a tertiary health care setting in the city of Peshawar, Pakistan.

### Inclusion/exclusion criteria

The inclusion criteria comprised T2DM patients who were aged 18 years and above, either gender, and with a history of poor glycemic control, having HbA1c > 7% for the last year or more despite using dual OHAs (full doses of Sitagliptin and Metformin). The exclusion criteria comprised T2DM patients with severe cognitive impairments, exposed to multiple drug combination before, and those not willing to participate in this study.

### Process of measurement

The patients were approached and the purpose of the study was explained to them in the presence of a consultant endocrinologist. From those who were willing to participate, informed written consent was obtained. Blood samples from the patients were collected in heparinized tubes, unless these were not available at that specific time, in which case the samples were collected in non-heparinized tubes. The non-heparinized tube samples were immediately centrifuged for 20 minutes at 2000 rpm. A clear serum supernatant was then used to evaluate various biochemical diagnostic factors, including serum creatinine, HbA1c, random blood sugar (RBS), fasting lipid profile (HDL, LDL, Triglycerides and cholesterol), urea, and fasting blood sugar (FBS).

### Study procedure

The demographic characteristics of the patients were recorded, including age, gender, weight, BMI, geographic location, comorbidities and biochemical diagnostic factors including HbA1c, fasting blood sugar (FBS), random blood sugar (RBS), lipid profile (HDL, LDL, Triglycerides, cholesterol), urea.

After meeting the inclusion criteria, the patients were randomly assigned into two groups (1 and 2).

**Group 1.** This cohort comprised a total of 40 T2DM patients. These patients were already on dual OHAs (Sitagliptin and Metformin), and a third agent (Empagliflozin, Glimepiride, or Gliclazide) was added to their existing therapy. Patients were advised to take the Sitagliptin + Metformin combination tablet twice a day, either with a meal or right after eating. Empagliflozin, Glimepiride and Gliclazide should be taken once in the morning, along with breakfast.

**Group 2.** This cohort also comprised a total of 40 T2DM patients. For these patients, pre-mixed Insulin (70/30) was added to their existing dual OHA therapy of Sitagliptin and Metformin.

After a 3-month interval, all laboratory parameters were repeated and recorded to assess the effectiveness of triple OHAs, and dual OHAs with Insulin, looking at glycemic control by comparing HbA1c levels at the baseline and endpoint.

### Ethics and dissemination

The study received approval from the Ethics Committee of Abdul Wali Khan University, Mardan, Pakistan. All procedures involving human participants adhered to the ethical standards set forth by this committee and were in accordance with the principles outlined in the 1964 Helsinki Declaration. Informed written consent was obtained from all patients who voluntarily participated in the study, with the understanding that they retained the right to withdraw at any point. Each patient was assigned a unique patient identity number for future reference. To minimize any bias, all data were treated with utmost confidentiality to prevent disclosure.

### Sample size

The study sample size was calculated based on previous studies for detecting the difference of 1% reduction for HbA1c (effect size), with a standard deviation of 1.4%, at 3 months in the intervention group [47]. A significance level of 0.05 is considered with a study power of 80%. Based on these numbers, the calculated sample size was 62 (31 for Cohort 1 and 31 for Cohort 2). To compensate for the 25% attrition rate, an additional 25% was added to the sample size. Overall, a total sample of 80 patients was assumed to be sufficient for this study.

### Statistical analysis

The data analysis was performed using SPSS version 22$^{®}$. For data that was categorical, frequency and percentages were reported, while for data that was continuous, mean and standard deviation were reported. In order to compare the pre- and post-effects of drugs to assess improvement in HbA1c and other laboratory parameters, a paired t-test was performed, while the comparison between two treatment groups were performed by using an independent t-test. A p-value <0.05 was considered statistically significant.

### Results

A total of n = 80 T2DM patients were included in this study (patients recruitment flow chart shown in **Fig 1**). Patients fulfilling the inclusion criteria were divided into two treatment groups. In group 1, 57.5% the patients were females and 47.5% were aged 41–55 years old, while in group 2, 52.5% were males and 60% of the patients were > 55 years old. Geographically, group 2 were mainly from Charsadda (52.5%) followed by 35% from Peshawar, whereas in group 2 the majority of patients (67.5%) were from Peshawar followed by 27.5% from Charsadda. Baseline weight and BMI distributions showed higher number of patients weighed 66–

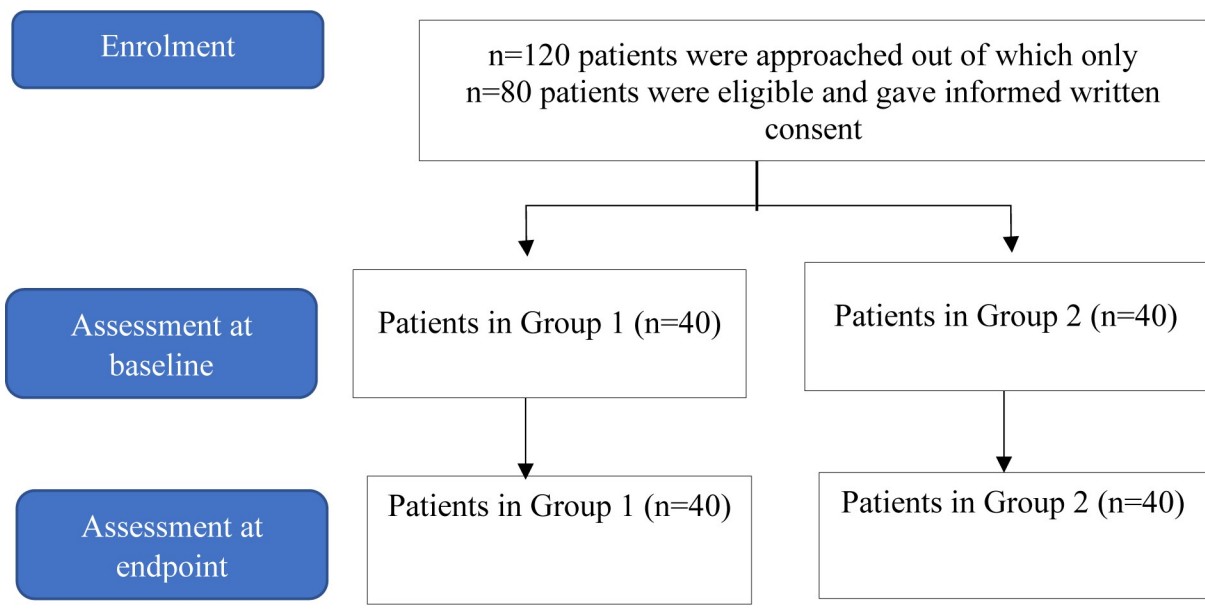

**Fig 1. Flow chart of patient's recruitment and participation.**

85 kg in both groups; while for BMI, 40% of the patients in group 1 had a BMI of 24.5–29.9, and in group 2, 35% had a BMI of 18.5–24.9. Both groups showed high baseline HbA1c levels: 50% of patients in group 1 had HbA1c >10, while in group 2, 57.5% of patients had HbA1c levels of 8.1–10, as shown in **Table 1**.

All the included patients in this study were on dual OHAs, i.e., Sitagliptin + Metformin. The patients were assigned into two groups. In group 1, the patients were started on an additional OHA (Empagliflozin, Gliclazide or Glimepiride) by their consultant endocrinologist. In group 2, patients were started on Insulin in addition to their dual OHAs.

**Table 2** shows the comparison of treatment groups based on patients' clinical profiles. In groups 1 and 2, HbA1c significantly decreased compared to baseline values (p = <0.001, and p = 0.002, respectively). In both the groups, weight, random and fasting blood sugar levels showed significant decreases at endpoint (p = <0.001, <0.001, and 0.006, respectively). BMI, cholesterol, and LDL significantly decreased among group 1 patients vs group 2 patients (p = <0.001 vs p = 0.131, p = 0.023 vs p = 0.896, and p = 0.003 vs p = 0.395, respectively). However, no significant change in triglycerides, HDL, urea, and serum creatinine were observed in either group.

**Fig 2** shows the impact of group 1 drugs on hyperglycemia based on HbA1c levels after 3 months of administration. At baseline, in the majority of the patients, HbA1c was poorly controlled (n = 38, 95%), whereas, after 3 months, the frequency of patients with poorly controlled diabetes decreased to 30 (75%), constituting a 20% improvement in HbA1c values. At baseline, none of the patients had HbA1c values lower than 7%; after 3 months of administering the study drugs, 10% of the patients had <7% HbA1c.

In group 2, HbA1c was poorly controlled in 95% of the patients at baseline, but after 3 months of treatment, the number of patients with poorly controlled diabetes decreased to 28 (70%), constituting a 25% improvement in HbA1c values. At baseline, none of the patients had controlled diabetes based on HbA1c values. 5% of patients achieved normal HbA1c values after 3 months of treatment, while in only one patient were HbA1c values < 7%.

**Table 1. Demographics and baseline characteristics of patients.**

| Variables | Treatment Groups | | | |
|---|---|---|---|---|
| | Group-1 | | Group-2 | |
| **Gender** | **n** | **%** | **n** | **%** |
| *Female* | 23 | 57.5 | 19 | 47.5 |
| *Male* | 17 | 42.5 | 21 | 52.5 |
| **Age (Years)** | | | | |
| *18–40* | 9 | 22.5 | 5 | 12.5 |
| *41–55* | 19 | 47.5 | 11 | 27.5 |
| *= >55* | 12 | 30 | 24 | 60 |
| **Address** | | | | |
| *Afghanistan* | 2 | 5 | 0 | 0 |
| *Bajaur* | 1 | 2.5 | 0 | 0 |
| *Charsadda* | 21 | 52.5 | 11 | 27.5 |
| *Dir Lower* | 0 | 0 | 1 | 2.5 |
| *Malakand* | 1 | 2.5 | 1 | 2.5 |
| *Peshawar* | 14 | 35 | 27 | 67.5 |
| *Swat* | 1 | 2.5 | 0 | 0 |
| **Baseline Weight (Kg)** | | | | |
| *35–65* | 10 | 25 | 14 | 35 |
| *66–85* | 22 | 55 | 18 | 45 |
| *>85* | 8 | 20 | 8 | 20 |
| **Baseline BMI** | | | | |
| *Below18.4* | 2 | 5 | 2 | 5 |
| *18.5–24.9* | 9 | 22.5 | 14 | 35 |
| *24.5–29.9* | 16 | 40 | 12 | 30 |
| *>29.9* | 13 | 32.5 | 12 | 30 |
| **Baseline HbA1c (%)** | | | | |
| *4–6.5* | 0 | 0 | 0 | 0 |
| *6.6–7* | 0 | 0 | 0 | 0 |
| *7.1–8* | 2 | 5 | 2 | 5 |
| *8.1–10* | 18 | 45 | 23 | 57.5 |
| *>10* | 20 | 50 | 15 | 37.5 |
| **Comorbidities** | | | | |
| *Chronic Kidney Disease* | 0 | 0 | 2 | 5 |
| *Diabetic Retinopathy* | 0 | 0 | 2 | 5 |
| *Hypertension* | 9 | 22.5 | 4 | 10 |
| *Ischemic Heart Disease* | 2 | 5 | 7 | 17.5 |
| *Ischemic stroke* | 0 | 0 | 1 | 2.5 |
| *No comorbidity* | 29 | 72.5 | 26 | 65 |

**Group 1:** Dual OHAs (Sitagliptin and Metformin), and a third agent (Empagliflozin, Glimepiride, or Gliclazide) was added to their existing therapy; **Group 2:** Insulin was added to their existing dual OHA therapy (Sitagliptin and Metformin).

The comparison between two treatment groups revealed significant differences. Patients in Group1 were younger the ages of patients, and had lower triglyceride and urea levels compared to group 2, with p-values of 0.039, and 0.033, respectively. Additionally, group 1 had lower HDL levels than group 2, with a p-value of 0.036, as shown in **Table 3**.

**Table 2. Comparison of study groups based on patients' variables before and after administration of study drugs.**

| Variables | Group-1 | | | | Group-2 | | | |
|---|---|---|---|---|---|---|---|---|
| | Pre | Post | Pre-Post | P-value | Pre | Post | Pre-Post | P-value |
| | Mean ± SD | Mean ± SD | % Diff | | Mean ± SD | Mean ± SD | %Diff | |
| *Weight (Kg)* | 75.1 ± 14.4 | 73.8 ± 13.9 | 1.74% | <**0.001**\* | 73.0 ± 14.9 | 71.8 ± 14.8 | 1.65% | <**0.001**\* |
| *Body Mass Index (BMI)* | 28.0 ± 5.1 | 27.5 ± 5.1 | 1.08% | <**0.001**\* | 27.3 ± 6.3 | 27.0 ± 6.2 | 1.10% | 0.131 |
| *HbA1c* | 10.3 ± 1.7 | 9.1 ± 1.6 | 12.37% | <**0.001**\* | 9.9 ± 1.6 | 9.0 ± 1.7 | 9.52% | **0.002**\* |
| *Random Blood Sugar (RBS)* | 282.6 ± 77.2 | 248.6 ± 77.3 | 12.80% | <**0.001**\* | 280.0 ± 79.7 | 258.3 ± 85.6 | 8.06% | **0.013**\* |
| *Fasting Blood Sugar (FBS)* | 181.4 ± 58.3 | 160.3 ± 52.3 | 12.35% | **0.006**\* | 190.8 ± 56.7 | 169.8 ± 58.6 | 11.64% | **0.005**\* |
| *Triglycerides* | 191.2 ± 36.1 | 184.9 ± 37.7 | 3.35% | 0.151 | 227.5 ± 106 | 224.2 ± 110.9 | 1.46% | 0.83 |
| *Cholesterol* | 188.0 ± 25.0 | 181.3 ± 24.9 | 3.62% | **0.023**\* | 181.3 ± 42.4 | 182.1 ± 40.2 | -0.44% | 0.896 |
| *Low-Density Lipoprotein (LDL)* | 110.8 ± 20.3 | 104.0 ± 23 | 6.33% | **0.003**\* | 103.5 ± 29.4 | 108.0 ± 26.3 | -4.25% | 0.395 |
| *High-Density Lipoprotein (HDL)* | 43.7 ± 2.8 | 44.6 ± 3.7 | -2.03% | 0.149 | 43.3 ± 8.8 | 42.8 ± 4 | 1.16% | 0.732 |
| *Serum Creatinine* | 0.9 ± 0.2 | 0.8 ± 0.2 | 11.76% | 0.878 | 1.3 ± 1.1 | 1.2 ± 1.2 | 8.0% | 0.303 |
| *Urea* | 32.0 ± 8 | 30.7 ± 6.4 | 4.14% | 0.34 | 38.9 ± 20.2 | 39.3 ± 24 | -1.02% | 0.885 |

Pair t-test was used, * p-value <0.05 was statistically significant

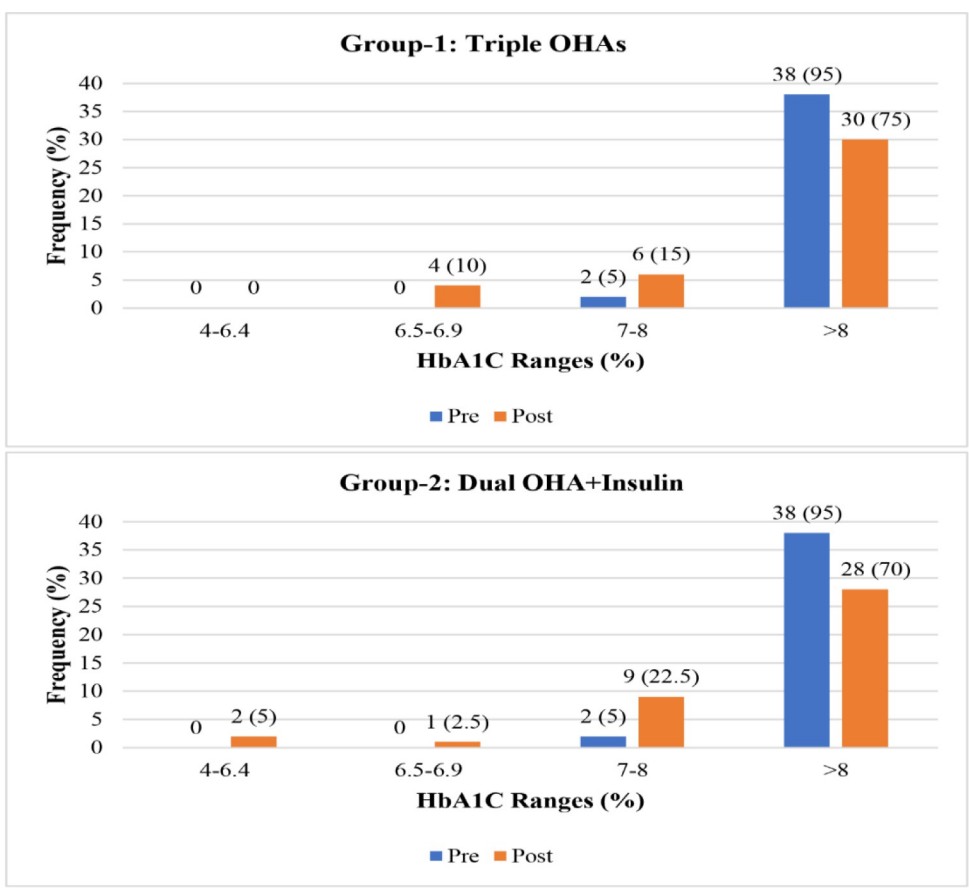

**Fig 2. Impact of group 1 and group 2 drugs on hyperglycemia based on HbA1c levels before and after 3 months of administration.**

**Table 3. Comparative analyses of group 1 vs group 2 based on various patient characteristics.**

| Variables | Group 1 | Group 2 | P-value |
|---|---|---|---|
| | Mean ± SD | Mean ± SD | |
| *Age (Years)* | 49.6 ± 12.6 | 56.9 ± 13.9 | **0.017*** |
| *Weight (Kg)* | 73.8 ± 13.9 | 71.8 ± 14.8 | 0.541 |
| *Body Mass Index (BMI)* | 27.5 ± 5.1 | 27.0 ± 6.2 | 0.683 |
| *HbA1c* | 9.1 ± 1.6 | 9.0 ± 1.70 | 0.724 |
| *Random Blood Sugar (RBS)* | 248.6 ± 77.3 | 258.3 ± 85.6 | 0.594 |
| *Fasting Blood Sugar (FBS)* | 160.3 ± 52.3 | 169.8 ± 58.6 | 0.443 |
| *Triglyceride* | 184.9 ± 37.7 | 224.2 ± 110.9 | **0.039*** |
| *Cholesterol* | 181.3 ± 24.9 | 182.1 ± 40.2 | 0.91 |
| *Low-Density Lipoprotein (LDL)* | 104.0 ± 23.0 | 108.0 ± 26.3 | 0.468 |
| *High-Density Lipoprotein (HDL)* | 44.6 ± 3.7 | 42.8 ± 40 | **0.036*** |
| *Urea* | 30.7 ± 6.4 | 39.3 ± 24 | **0.033*** |

An Independent T-test was used; a p-value <0.05 was considered statistically significant

The number of hypoglycemic episodes in group 1 patients was significantly less (n = 7; 7.5%) than in group 2 patients (p = 0.004) where 19 (47.5%) patients experienced hypoglycemic episodes, as shown in **Fig 3**.

## Discussion

T2DM is characterized by the gradual decline of beta-cell function and subsequent loss of glycemic control. When glycemic control is suboptimal, patients are at increased risk of developing microvascular and macrovascular complications [48]. Hence, it is crucial to identify treatment strategies that can effectively manage blood glucose levels, while also being convenient and adherable for patients.

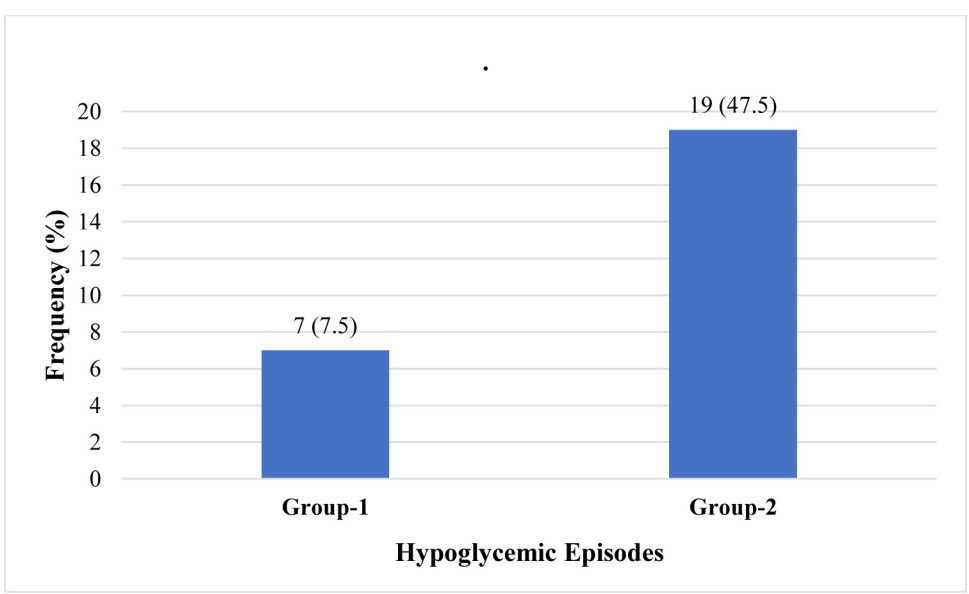

**Fig 3. Frequency of hypoglycemic episodes in group 1 vs group 2.**

Given the heightened prevalence of DM in Pakistan, along with a substantial number of patients experiencing uncontrolled DM, a significant portion of patients require triple therapy. The American Diabetes Association recommends the early initiation of Insulin, whereas commonly gliptins and SGLT-2 inhibitors are considered second-line treatment options. The addition of Insulin to oral regimens is a well-established approach that is effective for many patients [43]. Over the course of five years following the start of treatment, the incidence of secondary failure of OHAs in T2DM patients varies from 30% to 60% [49, 50]. The addition of a third agent, such as Insulin or an oral agent, is necessary for all patients with progressing T2DM in the future because the dual OHA regimen can only lower HbA1c levels by an average of 1.2–2.0% [51]. A study by Wallia et al. [52] reported that Insulin in combination with OHA is effective in achieving rapid glycemic control vs triple OHA regimens. Nevertheless, these guidelines were criticized for the fact that Insulin treatment can lead to frequent hypoglycemic episodes, weight gain, and limited patient satisfaction and compliance [53]. Furthermore, other studies reported the increased incidence of hypoglycemic episodes with Insulin [54–56]. Hence, this study aimed to compare the effectiveness of dual OHAs with additional Insulin versus triple OHAs in T2DM patients in improving HbA1c levels.

Our study found no statistically significant difference between the two treatment groups based on HbA1c, random blood sugar, and fasting blood sugar levels. These results are inconsistent with the findings of other studies which report significant differences between Insulin-based OHA regimens vs triple OHA regimens [53, 57]. Similarly, another study demonstrates that Insulin in combination with other OHA regimens provides better glycemic control [58], but at the cost of increased incidence of hypoglycemic episodes [53–56, 59]. These inconsistencies in results might be due to patients administering lower doses of Insulin or missing doses altogether out of concern for hypoglycemia [53, 60, 61], as our data reveals a notably higher occurrence of hypoglycemic episodes associated with Insulin use.

In this study, despite a triple OHA regimen, 4 (10%) of the patients in group 1 achieved less than 7% HbA1c values vs 3 (7.5%) in group 2 after 3 months of treatment, which is consistent with a study reporting one-fifth of patients being able to achieve HbA1c values lower than 7% [53]. Failure to achieve target HbA1c (<7%) may be attributed to non-compliance because of polypharmacy, as the majority of diabetic patients have comorbid illnesses (hypertension/ ischemic heart disease, dyslipidemia), meaning polypharmacy is inevitable, which compromises compliance to therapy for these patients [62]. Other barriers to achieving target HBA1c values include the long-term duration of diabetes, the severity of the disease, and the low effectiveness of some hypoglycemic drug classes [60, 61].

In the present study, group 1 patients showed a statistically significant reduction in triglycerides, HDL, and urea levels vs group 2 patients. Furthermore, patients in group 2 experienced a significantly higher number of hypoglycemic episodes compared to patients in group 1. These results are consistent with the findings of other studies where a higher incidence of hypoglycemic events was observed with Insulin-based regimens [53–56, 59].

An increase in body weight is one of the major side effects of Insulin therapy. A notable reduction in body weight was seen among patients of both groups. In group 1, a 1.3 kg reduction in body weight was seen after 3 months of treatment compared to baseline, whereas in group 2 patients, a 1.2 kg reduction in body weight was seen. These reductions in weight are consistent with the findings of other studies and may be attributed to Metformin which can be used in both non-obese and obese patients as a single agent or in combination for weight loss [63, 64]. Similarly, in Group 1 the weight reduction may be attributed to Empagliflozin, as 20% of patients were using Empagliflozin as a third OHA, which is consistent with the findings of another study [65]. However, we found no significant difference between the two groups

based on body weight, which is inconsistent with the findings of other studies where weight gain is reported with Insulin [53, 66].

Despite numerous published studies, the dilemma of finding an optimal therapeutic option following the failure of two OHA regimens in T2DM patients still exists. Previous research has demonstrated that Insulin in combination with Metformin is a viable, short-term treatment option that offers both safety and effectiveness in achieving rapid glycemic control, but at the cost of increased hypoglycemic episodes. Nevertheless, we did not find any significant difference based on glycemic control between triple OHAs (group 1) and dual OHAs plus Insulin (group 2). Hence, we suggest triple therapy as the effective treatment option for these patients based on the following points. Firstly, there is poor adherence to Insulin therapy: >50% of patients fail to continue Insulin therapy as prescribed by their physician, approximately 60% of patients intentionally skip Insulin injections, and on top of it all, fear of injection and patient stigma are still big hurdles [67]. Secondly, weight gain [66], hypoglycemic episodes [53], and the higher cost of Insulin are major deterrents to the lifelong use of Insulin-based therapies, especially in developing countries where the majority of patients have few financial resources [68]. Similarly, triple therapy led to a significant reduction in triglycerides, and a notable increase in HDL levels compared to Insulin-based therapy.

The study has several limitations. Firstly, the duration of 3 months was short. Secondly, the third OHA with Sitagliptin and Metformin in group 1 was not uniform. Thirdly, the study was conducted in a single center; incorporating patients from different centers might have resulted in more reliable results. Lastly, data on the duration of diabetes was not collected during data collection and it was realized after analyzing the data that the patients in group 2 were significantly older.

Overall, this study demonstrates the promising role of triple therapy vs dual therapy plus Insulin for glycemic control among T2DM patients. However, these findings must be treated with caution, since the sample size and follow-up duration of this study was small, and adherence to therapy was also not ascertained. Therefore, before these results may be endorsed, stringent clinical studies evaluating the long-term effects of these therapies, including therapy adherence, must be conducted.

## Conclusion

This study found no significant difference between a triple OHA regimen and a dual OHA plus Insulin regimen based on glycemic control. However, triple therapy may be preferable due to its lower incidence of hypoglycemic episodes, lower cost, and improvement in dyslipidemia compared to dual OHA plus Insulin therapy. It is important to note that further clinical research with rigorous designs is highly recommended to confirm these findings.

## Author Contributions

**Conceptualization:** Nadia Gul, Inayat Ur Rehman, Yasar shah, Long Chiau Ming.

**Data curation:** Nadia Gul, Khang Wen Goh.

**Formal analysis:** Nadia Gul, Inayat Ur Rehman, Zahid Ali, Amal K. Suleiman.

**Funding acquisition:** Khang Wen Goh, Long Chiau Ming.

**Investigation:** Nadia Gul, Inayat Ur Rehman, Yasar shah, Arbab Muhammad Ali, Zahid Ali, Omer Shehzad, Khang Wen Goh.

**Methodology:** Nadia Gul, Inayat Ur Rehman, Yasar shah, Arbab Muhammad Ali, Omer Shehzad, Long Chiau Ming, Amal K. Suleiman.

**Project administration:** Inayat Ur Rehman, Khang Wen Goh, Amal K. Suleiman.

**Software:** Zahid Ali, Long Chiau Ming.

**Supervision:** Inayat Ur Rehman, Yasar shah, Omer Shehzad.

**Validation:** Arbab Muhammad Ali, Khang Wen Goh, Long Chiau Ming, Amal K. Suleiman.

**Visualization:** Arbab Muhammad Ali.

**Writing – original draft:** Nadia Gul, Arbab Muhammad Ali, Zahid Ali.

**Writing – review & editing:** Inayat Ur Rehman, Yasar shah, Omer Shehzad, Khang Wen Goh, Long Chiau Ming, Amal K. Suleiman.

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
