## [Decision Letter · Decision Letter 0]

23 Jun 2024

PONE-D-24-14159Effectiveness of Dual Oral Hypoglycemic Agents plus Insulin versus Triple Oral Hypoglycemic Agents in Uncontrolled Type 2 Diabetes: A Pre- and Post-AnalysisPLOS ONE

Dear Dr. Rehman,

Thank you for submitting your manuscript to PLOS ONE. After careful consideration, we feel that it has merit but does not fully meet PLOS ONE’s publication criteria as it currently stands. Therefore, we invite you to submit a revised version of the manuscript that addresses the points raised during the review process.

We look forward to receiving your revised manuscript.

Kind regards,

Nimesh Lageju

Academic Editor

PLOS ONE

2. In the online submission form, you indicated that [Data will be made available on request to inayat.rehman@awkum.edu.pk.]. 

Additional Editor Comments (if provided):

Reviewers' comments:

Reviewer's Responses to Questions

**Comments to the Author**

1. Is the manuscript technically sound, and do the data support the conclusions?

Reviewer #1: Partly

Reviewer #2: Partly

Reviewer #3: Partly

2. Has the statistical analysis been performed appropriately and rigorously? 

Reviewer #1: Yes

Reviewer #2: Yes

Reviewer #3: Yes

3. Have the authors made all data underlying the findings in their manuscript fully available?

Reviewer #1: Yes

Reviewer #2: Yes

Reviewer #3: Yes

4. Is the manuscript presented in an intelligible fashion and written in standard English?

Reviewer #1: No

Reviewer #2: Yes

Reviewer #3: Yes

5. Review Comments to the Author

Reviewer #1: The authors address a subject of crucial importance as Diabetes mellitus is increasing its prevalence each year. Despite the variety of drugs and insulin, it's still a challenge to decide the best regimen for better glycemic control and reduce its complications. Although a few studies suggest that starting with combined OHA may achieve better results, others suggest that the better approach is a shared decision with the patient; there isn't a definitive study to show whether adding a third OHA or insulin to achieve satisfactory glycemic control is better. Although a pre-post study design is less expensive than the RCT, does not require randomization, and can provide some information about the effect of the intervention alongside its use, it has some limitations as it cannot avoid bias either state for sure if the outcome can be attributed to the procedure. Some other points need to be clarified, mainly related to the methods.

#1 In the abstract, some methods are described in the results.

#2 Page 4, line 105: The item inclusion/exclusion criteria don't say that only patients using sitagliptin+ metformin would be included or whether there is a protocol that patients use these drugs as the first line of treatment. It would be helpful to determine whether the patients were in total doses of the medications when they started the third OHD or insulin. And explain how they began and titrated the drug or insulin dose.

#3Page 5-6, Table 1: I would not expect that Table 1 shows the baseline treatment; all patients in group 1 were already on a 3-drug regimen. Was that a mistake?

#4 In the results, it would be helpful to include, if available, the data on dropouts or compliance with the treatment.

#5 From page 10 to the end, the lines are not numbered

#6 Page 11, in the discussion, what do you mean by...' despite aggressive regimens'?

#7 On page 11, the authors attribute the weight loss to metformin. Is there any other data to support that? All patients were using metformin at the baseline, and 20% of group 1 started taking Empagliflozin.

#8 The manuscript needs corrections in the grammar and punctuation.

Reviewer #2: Review Comments

1. On the scope and title: The title is too long and needs adequate shortening. It should be guided by the scope of the study. Commonly we write as “type II Diabetes Mellitus” not “type 2 Diabetes Mellitus”.

2. On the abstract: The abstract lacks clarity and logical presentation E.g., the type of study design and analysis is missed. The result and the conclusion are inconsistent.

3. On the background: It is little bit far from the study objective. E.g., Very few points are described regarding the treatment of uncontrolled type II Diabetes and its outcome. In addition, it contains mixed of ideas and lacks linking between points. E.g., It directly jumps from global prevalence to Pakistan. Most importantly, the section lacks the main problem statement of the study.

4. On the methods section: Similarly, the methods section is incomplete for the study design, process of measurements, the sampling procedure and selection of participants as well as the type of analysis. Ethics should be presented in detail in line with the protocol of the study.

5. On the result: it lacks logical flow and brief presentation including lack of self-explanatory of tables.

6. The discussion is inadequate and needs rich explanation.

7. Others: In addition, lacks consistency E.g., you have used inclusion criteria in line 105-109 and ‘illegibility criteria’ in line 115.

Regards,

Reviewer #3: Thank you, Editor, for inviting me to review this research paper.

Authors, your research paper provides valuable insights in the field. However, some aspects need clarification and additional details to enhance the comprehensibility and robustness of your work. Below are my comments and suggestions:

Abstract:

1. Aim: Please rewrite the aim of the study to be clearer.

2. Glycemic Episodes: In the results section of the abstract, add the p-value (p=0.004) at the end of the sentence discussing glycemic episodes for better clarity.

Introduction:

3. Line 71: It would be interesting to include the prevalence of glycemic episodes among men and women in Pakistan to provide context and relevance to your study.

Methods:

4. Rationale for Criteria: Explain why you chose those with HbA1c more than 7 as the threshold. What is the rationale behind this choice? Educating the readers on this decision would enhance understanding.

5. Participant Assignment: Clarify how participants were assigned to group 1 and group 2. Were they randomly allocated? This information is crucial for assessing the validity of your study design.

6. Study Awareness: Describe how patients were informed about the study. This could include recruitment strategies.

7. Literature Support: Include literature that supports the notion that three months is a sufficient period to observe changes in patients' HBA1c levels.

8. Data Distribution: Indicate whether your data were normally distributed. This affects the choice of statistical tests and the interpretation of results.

Results:

9. Terminology: On line 157, change "the majority" were females to reflect the actual percentage. Referring to 52% as a majority is misleading. Similarly, clarify that 50% does not constitute the majority.

10. Follow-Up: Specify if there were any losses to follow-up. If yes, provide the number of patients lost and discuss the potential impact on your results.

11. Compliance: Discuss how you ensured patients' compliance with the treatment regimen. How did you confirm that they took their medications on time? I am aware that the authors mentioned this as a limitation, but did the researchers not address patient compliance at all?

12. Group Allocation: Reiterate how patients were allocated to each group. This should be clear and transparent to readers.

13. Rationale for Grouping: Explain the rationale behind your age groups, weight, and BMI categories. Providing a justification for these groupings would help in understanding your results.

14. Table 1: The oral glycemic agents in Table 1 are confusing. Please present this information more clearly.

Discussion:

15. HbA1c Values: Refer to HbA1c as a value and not a percentage.

16. Study Limitations: Given the study limitations, particularly the lack of tracking patient compliance, consider adding "A Pilot Study" to your title to accurately reflect the scope and preliminary nature of your research.

Your research is valuable, and with these revisions, it can provide even more robust and clear insights. I look forward to seeing the revised version.

6. PLOS authors have the option to publish the peer review history of their article (what does this mean?). If published, this will include your full peer review and any attached files.

Reviewer #1: No

Reviewer #2: No

Reviewer #3: **Yes: **Mona Abdelrehim

---

## [Author Response · Author response to Decision Letter 0]

10 Aug 2024

Dear Editor and Reviewers,

Thank you for taking the time to provide such detailed and helpful comments on our manuscript. We appreciate your constructive criticisms and suggestions, and we believe they will significantly improve the quality of our manuscript. We have revised the manuscript based on your feedback and would like to provide the following responses to your concerns.

Reviewer #1:

The authors address a subject of crucial importance as Diabetes mellitus is increasing its prevalence each year. Despite the variety of drugs and insulin, it's still a challenge to decide the best regimen for better glycemic control and reduce its complications. Although a few studies suggest that starting with combined OHA may achieve better results, others suggest that the better approach is a shared decision with the patient; there isn't a definitive study to show whether adding a third OHA or insulin to achieve satisfactory glycemic control is better. Although a pre-post study design is less expensive than the RCT, does not require randomization, and can provide some information about the effect of the intervention alongside its use, it has some limitations as it cannot avoid bias either state for sure if the outcome can be attributed to the procedure. Some other points need to be clarified, mainly related to the methods.

#1 In the abstract, some methods are described in the results.

Response: Thank you for the correction, suggested changes are done in the manuscript. 

#2 Page 4, line 105: The item inclusion/exclusion criteria don't say that only patients using sitagliptin+ metformin would be included or whether there is a protocol that patients use these drugs as the first line of treatment. It would be helpful to determine whether the patients were in total doses of the medications when they started the third OHD or insulin. And explain how they began and titrated the drug or insulin dose.

Response: Thank you for the comment, the inclusion criteria comprised of patients who were on full doses of sitagliptin+ metformin as a first line therapy and their diabetes was uncontrolled based on HbA1c values. The suggested changes have been done.

#3Page 5-6, Table 1: I would not expect that Table 1 shows the baseline treatment; all patients in group 1 were already on a 3-drug regimen. Was that a mistake?

Response: Thank you for the highlight this issue. This was a mistake which was overlooked at time of submission. The correction has been made in the manuscript and the details are removed from Table 1 to avoid confusion to readers. The details of group 1 and group 2 are given in text after table 1.

#4 In the results, it would be helpful to include, if available, the data on dropouts or compliance with the treatment.

Response: Thank you for comment, the compliance was ensured by principal author as the medication were given to patients free of cost and on monthly basis the patients were instructed to meet the principal author to get the next month free doses of their medication. This also help to avoid the drop out of the patients in the study. 

#5 From page 10 to the end, the lines are not numbered

Response: Thank you for comment, the line numbers are assigned to the whole manuscript as suggested. 

#6 Page 11, in the discussion, what do you mean by...' despite aggressive regimens?

Response: Thank you for comment and highlighting this word, to avoid confusion to readers this has been changes to “triple OHAs regimen”. 

#7 On page 11, the authors attribute the weight loss to metformin. Is there any other data to support that? All patients were using metformin at the baseline, and 20% of group 1 started taking Empagliflozin.

Response: Thank you for comment, the statement has been modified and updated to “Similarly, in Group 1 the weight reduction may be attributed to empagliflozin as 20% of the patients were using empagliflozin as a third OHAs, which are consistent with findings of another study [51]”.

#8 The manuscript needs corrections in the grammar and punctuation.

Response: Thank you for comment, the whole manuscript has been thoroughly edited by native English speaker. 

Reviewer #2:

1. On the scope and title: The title is too long and needs adequate shortening. It should be guided by the scope of the study. Commonly we write as “type II Diabetes Mellitus” not “type 2 Diabetes Mellitus”.

Response: Thank you for comment, the title is shorten as suggested to “Comparing Dual Oral Agents Plus Insulin vs. Triple Oral Agents in Uncontrolled Type II Diabetes: A Pilot Study” and the “type 2 Diabetes Mellitus” is replaced by “type II Diabetes Mellitus” in the manuscript. We included “pilot study” in title as it is suggested by one of the reviewers. 

2. On the abstract: The abstract lacks clarity and logical presentation E.g., the type of study design and analysis is missed. The result and the conclusion are inconsistent.

Response: Thank you for the correction, suggested changes are done in the manuscript. The results and conclusion were rewritten again as suggested. 

3. On the background: It is little bit far from the study objective. E.g., Very few points are described regarding the treatment of uncontrolled type II Diabetes and its outcome. In addition, it contains mixed of ideas and lacks linking between points. E.g., It directly jumps from global prevalence to Pakistan. Most importantly, the section lacks the main problem statement of the study.

Response: Thank you for comment, the background section has been updated as suggested.

4. On the methods section: Similarly, the methods section is incomplete for the study design, process of measurements, the sampling procedure and selection of participants as well as the type of analysis. Ethics should be presented in detail in line with the protocol of the study.

Response: Thank you for comment, the method section has been updated as suggested. However, the ethics approval details are not mentioned as the journal requested us to remove these details at the time of peer-review process.

5. On the result: it lacks logical flow and brief presentation including lack of self-explanatory of tables.

Response: Thank you for the comment, the result section is rewritten again to avoid confusion to the readers. 

6. The discussion is inadequate and needs rich explanation.

Response: Thank you for the comment, the discussion section is updated as suggested.

7. Others: In addition, lacks consistency E.g., you have used inclusion criteria in line 105-109 and ‘illegibility criteria’ in line 115.

Response: Thank you for the highlight, the correction is done in manuscript as suggested. 

Reviewer #3:

Thank you, Editor, for inviting me to review this research paper. Authors, your research paper provides valuable insights in the field. However, some aspects need clarification and additional details to enhance the comprehensibility and robustness of your work. Below are my comments and suggestions:

Abstract:

1. Aim: Please rewrite the aim of the study to be clearer.

Response: Thank you for comment, the aim is rewritten again as suggested.

2. Glycemic Episodes: In the results section of the abstract, add the p-value (p=0.004) at the end of the sentence discussing glycemic episodes for better clarity.

Response: Thank you for comment, the suggested changes has been done.

Introduction:

3. Line 71: It would be interesting to include the prevalence of glycemic episodes among men and women in Pakistan to provide context and relevance to your study.

Response: Thank you for comment, the suggested changes has been done.

Methods:

4. Rationale for Criteria: Explain why you chose those with HbA1c more than 7 as the threshold. What is the rationale behind this choice? Educating the readers on this decision would enhance understanding.

Response: Thank you for comment, the laboratory standard use in this study showed a HbA1c range: Normal: 4-6.5, Pre-diabetes: 6.6-7, diabetes: >7.1-8.0. we used this criterion and this was the rationale behind this choice. 

5. Participant Assignment: Clarify how participants were assigned to group 1 and group 2. Were they randomly allocated? This information is crucial for assessing the validity of your study design.

Response: Thank you for comment, the statement is updated in the manuscript as below: “After meeting the inclusion criteria, the patients were randomly assigned into two groups”.

6. Study Awareness: Describe how patients were informed about the study. This could include recruitment strategies.

Response: Thank you for comment, the statement is updated in the manuscript as below: “The patients were approached and the purpose of the study was explained to them in Prescence of consultant endocrinologist. Those who were willing to participate an informed written consent was obtained from them”. 

7. Literature Support: Include literature that supports the notion that three months is a sufficient period to observe changes in patients' HBA1c levels.

Response: Thank you for comment, the suggested changes has been done. “HbA1c is a marker of the average glucose levels spread over a two- to three-month period [25]”.

8. Data Distribution: Indicate whether your data were normally distributed. This affects the choice of statistical tests and the interpretation of results.

Response: Thank you for comment, the normality of the data was checked by using Shapiro-Wilk Test and the data showed normal distribution. Keeping in mind the normality of data, the data was presented as mean and SD. Parametric test were performed in this study as data was normally distributed. 

Results:

9. Terminology: On line 157, change "the majority" were females to reflect the actual percentage. Referring to 52% as a majority is misleading. Similarly, clarify that 50% does not constitute the majority.

Response: Thank you for comment, the result section is rewritten again to avoid confusion to readers.

10. Follow-Up: Specify if there were any losses to follow-up. If yes, provide the number of patients lost and discuss the potential impact on your results.

Response: Thank you for comment, there was no drop out in this study. The principal author was responsible for giving medication free of cost to the patients and on monthly basis the patients were instructed to meet the principal author to get the next month free doses of their medication. This also help to avoid the drop out of the patients in the study. 

11. Compliance: Discuss how you ensured patients' compliance with the treatment regimen. How did you confirm that they took their medications on time? I am aware that the authors mentioned this as a limitation, but did the researchers not address patient compliance at all?

Response: Thank you for comment, the principal author was responsible for giving medication free of cost to the patients and on monthly basis the patients were instructed to meet the principal author to get the next month free doses of their medication. This also helped in compliance of the patients toward their medication. However, no specific scale/instrument was utilized for evaluation of the adherence to therapy that’s why this was included as a limitation in limitation section. 

12. Group Allocation: Reiterate how patients were allocated to each group. This should be clear and transparent to readers.

Response: Thank you for comment, the statement is incorporated in the manuscript as: “After meeting the inclusion criteria, the patients were randomly assigned into two groups”

13. Rationale for Grouping: Explain the rationale behind your age groups, weight, and BMI categories. Providing a justification for these groupings would help in understanding your results.

Response: Thank you for comment, the age and weight were categorized based on the distribution of patient keeping in mind a comparable number of patients in each category. However, for the BMI, WHO classification was used. 

14. Table 1: The oral glycemic agents in Table 1 are confusing. Please present this information more clearly.

Response: Thank you for comment, the details are removed from table 1 and explained at end of table 1 as below: 

“All the included patients in this study were on dual OHAs i.e. Sitagliptin + Metformin. The patients were assigned into two groups. In group 1, the patients were started with additional third OHA i.e. Empagliflozin or Gliclazide or Glimepiride by their consultant endocrinologist. While for the group 2 patients were started with insulin in addition to their dual OHAs i.e. Sitagliptin + Metformin”.

Discussion:

15. HbA1c Values: Refer to HbA1c as a value and not a percentage.

Response: Thank you for comment, the HbA1c are generally presented as percentages rather than value. That is why they are presented in percentages. 

16. Study Limitations: Given the study limitations, particularly the lack of tracking patient compliance, consider adding "A Pilot Study" to your title to accurately reflect the scope and preliminary nature of your research.

Response: Thank you for comment, the suggested changes has been done. 

Again, thank you for your time and valuable feedback. We believe that the revisions have significantly improved the manuscript and hope that it is now suitable for publication.

Regard’s

Corresponding author

---

## [Decision Letter · Decision Letter 1]

9 Sep 2024

PONE-D-24-14159R1Comparing Dual Oral Agents Plus Insulin vs. Triple Oral Agents in Uncontrolled Type II Diabetes: A Pilot StudyPLOS ONE

Dear Dr. Rehman,

Thank you for submitting your manuscript to PLOS ONE. After careful consideration, we feel that it has merit but does not fully meet PLOS ONE’s publication criteria as it currently stands. Therefore, we invite you to submit a revised version of the manuscript that addresses the points raised during the review process.

We look forward to receiving your revised manuscript.

Kind regards,

Nimesh Lageju

Academic Editor

PLOS ONE

Journal Requirements:

Reviewers' comments:

Reviewer's Responses to Questions

**Comments to the Author**

1. If the authors have adequately addressed your comments raised in a previous round of review and you feel that this manuscript is now acceptable for publication, you may indicate that here to bypass the “Comments to the Author” section, enter your conflict of interest statement in the “Confidential to Editor” section, and submit your "Accept" recommendation.

Reviewer #4: All comments have been addressed

Reviewer #5: All comments have been addressed

2. Is the manuscript technically sound, and do the data support the conclusions?

Reviewer #4: Yes

Reviewer #5: Partly

3. Has the statistical analysis been performed appropriately and rigorously? 

Reviewer #4: Yes

Reviewer #5: I Don't Know

4. Have the authors made all data underlying the findings in their manuscript fully available?

Reviewer #4: Yes

Reviewer #5: Yes

5. Is the manuscript presented in an intelligible fashion and written in standard English?

Reviewer #4: Yes

Reviewer #5: Yes

6. Review Comments to the Author

Reviewer #4: This study has some limitations, which I think authors have already acknowledged in the discussion section. The authors have also addressed the majority of the other reviewers' comments. However, I have the following additional comments/suggestions to authors:

1. For ease of understanding, authors should consider including a flow chart that shows the number of participants evaluated for potential enrolment into the trial and the number excluded because they did not meet the inclusion criteria or declined to participate.

2. Authors should also include information on the time of the day when treatment was given to participants, as well as whether the drugs were given before or after a meal. This information can be added to the methods/results section.

3. Figure legends should be expanded to include brief descriptions of the plots depicted.

Reviewer #5: Thank you for offering me an invitation to be the reviewer of this manuscript “Comparing Dual Oral Agents Plus Insulin vs. Triple Oral Agents in Uncontrolled Type II Diabetes: A Pilot Study”

The authors had submitted a manuscript comparing dual OHA’s with Insulin and triple OHA’s in uncontrolled Type II diabetes. The selected topic becomes increasingly significant given the burden of diabetes in Asian countries.

I am giving my observations and comments for the authors to consider:

I observe the corresponding author in the first page is different from the one mentioned in page 1 line 18-24

Introduction: The introduction effectively justifies the chosen topic and the methods with convincing arguments supported by relevant references, as required.

Line 41: Insulin (please maintain the Case)

Page line 62-63: Type 1 DM increases Type 2 DM prevalence? May be rewritten as “There is an increasing trend in Type 1 DM worldwide as well”.

Line 66-67: Repeat info, may be removed

Line 70-71 Ref 16 – IDF Diabetes Atlas 10th edition may be a better reference for this information

Line 79 - Instead of “ T2DM” uncontrolled T2 DM may be used

Line 81-82: mention as “Disease” rather than “Damage”

Line 78-85: May be made briefer and to the point using scientific terms

Line 94-98 may be moved down after discussion on preferred dual therapy.

Justification may be given for choosing patients on Sitagliptin & Metformin combination, is that the preferred dual therapy in the geographical area?

Inclusion/Exclusion Criteria: It will be helpful to know if patients who are exposed to multiple drug combination before, are also excluded.

Study Procedure:

Line 138: Term “renal functioning test” may be removed

Line 141: It will be good to know how the third drug is chosen between the three choices

Line 147: Grp 2: Please mention the type of Insulin added - analogs, rapid or long acting or mix, etc

Results:

Table1: The information regarding drug regimen is missing. This information may help to explain the weight gain and glycemic control results. Suggesting to add.

Data on diabetes age of the patients if available will be significant to interpret the results considering the fact that grp 2 patients are significantly older.

Figure 1:

Group1 showed 20% improvement in glycemic control while in Group 2, 30 % of participants showed improvement. The reduction in HbA1c depends on the baseline level. Since number of patients with HbA1c more than 10 are considerable in both the groups, data of ‘more than 8’ may be further grouped and analysed.

Line 220-221: Higher the HDL is a positive outcome. Suggested to rewrite the sentence

Line 240-241: Is this an error? that patients prefer Insulin over OHA? If it is an error, please correct.

The limitations of the study are well-documented and enumerated.

7. PLOS authors have the option to publish the peer review history of their article (what does this mean?). If published, this will include your full peer review and any attached files.

Reviewer #4: **Yes: **Suhail A. Ansari

Reviewer #5: No

---

## [Author Response · Author response to Decision Letter 1]

16 Sep 2024

Dear Editor and Reviewers,

Thank you for taking the time to provide such detailed and helpful comments on our manuscript. We appreciate your constructive criticisms and suggestions, and we believe they will significantly improve the quality of our manuscript. We have revised the manuscript based on your feedback and would like to provide the following responses to your concerns.

Reply to comments

Reviewer #4

This study has some limitations, which I think authors have already acknowledged in the discussion section. The authors have also addressed the majority of the other reviewers' comments. However, I have the following additional comments/suggestions to authors:

Comment: 1. For ease of understanding, authors should consider including a flow chart that shows the number of participants evaluated for potential enrolment into the trial and the number excluded because they did not meet the inclusion criteria or declined to participate.

Response: Thank you for your comment, the flowchart is added to the manuscript as Figure 1.

Comment: 2. Authors should also include information on the time of the day when treatment was given to participants, as well as whether the drugs were given before or after a meal. This information can be added to the methods/results section.

Response: Thank you for your comment. Patients were advised to take the Sitagliptin + Metformin combination tablet twice a day, either with a meal or right after eating. Empagliflozin should be taken once in the morning, along with breakfast. Glimepiride is also to be taken once daily, with breakfast. Similarly, Gliclazide is taken once daily in the morning, with a meal.

Comment: 3. Figure legends should be expanded to include brief descriptions of the plots depicted.

Response: Thank you for your comment, the suggested changes have been incorporated in the manuscript.

Reviewer #5

Thank you for offering me an invitation to be the reviewer of this manuscript “Comparing Dual Oral Agents Plus Insulin vs. Triple Oral Agents in Uncontrolled Type II Diabetes: A Pilot Study”. The authors had submitted a manuscript comparing dual OHA’s with Insulin and triple OHA’s in uncontrolled Type II diabetes. The selected topic becomes increasingly significant given the burden of diabetes in Asian countries.

I am giving my observations and comments for the authors to consider:

Comment: I observe the corresponding author in the first page is different from the one mentioned in page 1 line 18-24.

Response: Thank you for your constructive comment, the corresponding author in the manuscript is to be considered while ignoring the submission author.

Introduction: The introduction effectively justifies the chosen topic and the methods with convincing arguments supported by relevant references, as required.

Comment: Line 41: Insulin (please maintain the Case)

Response: Thank you for your comment, the suggested changes have been made to the manuscript.

Comment: Page line 62-63: Type 1 DM increases Type 2 DM prevalence? May be rewritten as “There is an increasing trend in Type 1 DM worldwide as well”.

Response: Thank you for your comment, the suggested changes have been made to the manuscript.

Comment: Line 66-67: Repeat info, may be removed

Response: Thank you for your comment, the suggested changes have been made to the manuscript.

Comment: Line 70-71 Ref 16 – IDF Diabetes Atlas 10th edition may be a better reference for this information

Response: Thank you for your comment, the suggested changes have been made in the manuscript to IDF Diabetes Atlas 10th Edition in Reference 16.

Comment: Line 79 - Instead of “T2DM” uncontrolled T2 DM may be used

Response: Thank you for your comment, the suggested changes have been made to the manuscript as per the below comment to make it concise using scientific terms.

Comment: Line 81-82: mention as “Disease” rather than “Damage”

Response: Thank you for your comment, the suggested changes have been made to the manuscript.

Comment: Line 78-85: May be made briefer and to the point using scientific terms

Response: Thank you for your comment, the suggested changes have been made to the manuscript.

Comment: Line 94-98 may be moved down after discussion on preferred dual therapy.

Response: Thank you for the comment, the section is moved to page no 8, discussion section after the preferred dual therapy as suggested.

Comment: Justification may be given for choosing patients on Sitagliptin & Metformin combination, is that the preferred dual therapy in the geographical area?

 Response: Thank you for your comment, Sitagliptin & Metformin combination is the preferred dual therapy is the preferred choice by endocrinologists in our geographic location. 

Comment: Inclusion/Exclusion Criteria: It will be helpful to know if patients who are exposed to multiple drug combination before, are also excluded.

Response: Thank you for your comment, the suggested changes have been made to the relevant section i.e. inclusion/exclusion criteria in the manuscript.

Study Procedure:

Comment: Line 138: Term “renal functioning test” may be removed

Response: Thank you for your comment, the suggested text is removed from the manuscript.

Comment: Line 141: It will be good to know how the third drug is chosen between the three choices.

Response: Thank you for your comment, Empagliflozin and Gliclazide were added to the other two oral anti-diabetic medications because both offer cardiovascular benefits. Empagliflozin also has kidney-protective effects, which is important since many patients in Group 1 had high blood pressure and heart disease. Similarly, Glimepiride was chosen for its ability to slow the progression of diabetes and reduce the need for insulin, while also being cost-effective. The choice of a third medication to add to the existing two was based on the patient's overall health, including comorbid conditions, and sometimes the endocrinologist's preference for a specific medication.

Comment: Line 147: Grp 2: Please mention the type of Insulin added - analogs, rapid or long acting or mix, etc.

Response: Thank you for your comment, the Insulin used in our study was pre-mixed Insulin (70/30), and the same has been added in the method section under the section Group 2 heading. 

Results:

Comment: Table1: The information regarding drug regimen is missing. This information may help to explain the weight gain and glycemic control results. Suggesting to add.

Response: Thank you for highlighting this; to avoid confusion, a footnote is added at the end of table 1 showing Group 1 and Group 2 along with details of the treatment regimen. 

Comment: Data on diabetes age of the patients if available will be significant to interpret the results considering the fact that grp 2 patients are significantly older.

Response: Thank you for highlighting this point, the data on how long patients had diabetes was not collected and this could be a possible limitation of this study which is added to the limitation section of the manuscript.

Comment: Figure 1: Group1 showed 20% improvement in glycemic control while in Group 2, 30 % of participants showed improvement. The reduction in HbA1c depends on the baseline level. Since number of patients with HbA1c more than 10 are considerable in both the groups, data of ‘more than 8’ may be further grouped and analysed.

Response: Thank you for your comment, the patients recruited in the study were having HbA1c > 7% for the last year or more despite using dual OHAs (full doses of Sitagliptin and Metformin). As suggested the data was checked again for further analysis while grouping them in having HbA1c more than 8, the data revealed that only 1 participant/patient in each group was having HbA1c less than 8 i.e. 7.9 in both groups. Due to this further group and analysis will have no impact on the findings. I am hopeful that the honorable reviewer will agree with our point of view. 

Comment: Line 220-221: Higher the HDL is a positive outcome. Suggested to rewrite the sentence

Response: Thank you for your comment, the suggested changes have been made to the manuscript.

Comment: Line 240-241: Is this an error? that patients prefer Insulin over OHA? If it is an error, please correct. 

Response: Thank you for the comment, that was a typo and is removed from the manuscript to avoid confusion to the readers.

Comment: The limitations of the study are well-documented and enumerated.

Response: Thank you for your comment, we are thankful for sparing time for a thorough review of the manuscript. 

Again, thank you for your time and valuable feedback. We believe that the revisions have significantly improved the manuscript and hope that it is now suitable for publication.

Regards

Corresponding author

---

## [Editor Report · Decision Letter 2]

20 Sep 2024

Comparing Dual Oral Agents Plus Insulin vs. Triple Oral Agents in Uncontrolled Type II Diabetes: A Pilot Study

PONE-D-24-14159R2

Dear Dr. Rehman,

We’re pleased to inform you that your manuscript has been judged scientifically suitable for publication and will be formally accepted for publication once it meets all outstanding technical requirements.

Kind regards,

Nimesh Lageju

Academic Editor

PLOS ONE
---

## [Editor Report · Acceptance letter]

12 Nov 2024

PONE-D-24-14159R2 

PLOS ONE

Dear Dr. Rehman, 

I'm pleased to inform you that your manuscript has been deemed suitable for publication in PLOS ONE. Congratulations! Your manuscript is now being handed over to our production team.

Kind regards, 

on behalf of

Dr. Nimesh Lageju 

Academic Editor

PLOS ONE